# Detection of H3K4me3 Identifies NeuroHIV Signatures, Genomic Effects of Methamphetamine and Addiction Pathways in Postmortem HIV+ Brain Specimens that Are Not Amenable to Transcriptome Analysis

**DOI:** 10.3390/v13040544

**Published:** 2021-03-24

**Authors:** Liana Basova, Alexander Lindsey, Anne Marie McGovern, Ronald J. Ellis, Maria Cecilia Garibaldi Marcondes

**Affiliations:** 1San Diego Biomedical Research Institute, San Diego, CA 92121, USA; lbasova@sdbri.org (L.B.); alindsey@SDBRI.ORG (A.L.); annemariermcgovern@gmail.com (A.M.M.); 2Departments of Neurosciences and Psychiatry, University of California San Diego, San Diego, CA 92103, USA; roellis@health.ucsd.edu

**Keywords:** postmortem interval, HIV, methamphetamine, brain, H3K4me3

## Abstract

Human postmortem specimens are extremely valuable resources for investigating translational hypotheses. Tissue repositories collect clinically assessed specimens from people with and without HIV, including age, viral load, treatments, substance use patterns and cognitive functions. One challenge is the limited number of specimens suitable for transcriptional studies, mainly due to poor RNA quality resulting from long postmortem intervals. We hypothesized that epigenomic signatures would be more stable than RNA for assessing global changes associated with outcomes of interest. We found that H3K27Ac or RNA Polymerase (Pol) were not consistently detected by Chromatin Immunoprecipitation (ChIP), while the enhancer H3K4me3 histone modification was abundant and stable up to the 72 h postmortem. We tested our ability to use H3K4me3 in human prefrontal cortex from HIV+ individuals meeting criteria for methamphetamine use disorder or not (Meth +/−) which exhibited poor RNA quality and were not suitable for transcriptional profiling. Systems strategies that are typically used in transcriptional metadata were applied to H3K4me3 peaks revealing consistent genomic activity differences in regions where addiction and neuronal synapses pathway genes are represented, including genes of the dopaminergic system, as well as inflammatory pathways. The resulting comparisons mirrored previously observed effects of Meth on suppressing gene expression and provided insights on neurological processes affected by Meth. The results suggested that H3K4me3 detection in chromatin may reflect transcriptional patterns, thus providing opportunities for analysis of larger numbers of specimens from cases with substance use and neurological deficits. In conclusion, the detection of H3K4me3 in isolated chromatin can be an alternative to transcriptome strategies to increase the power of association using specimens with long postmortem intervals and low RNA quality.

## 1. Introduction

Postmortem human specimens represent an extremely valuable resource with direct translational implications. Tissue repositories collect a great number of brain specimens from HIV-positive (+) and negative individuals, with different viral loads, treatments and ages. Most importantly, specimens may also be from subjects with different substance use disorder patterns and cognitive functions, generating opportunities for identifying important molecular elements tied to recorded clinical and individual variables. Currently, the analysis of the transcriptome can provide in-depth information about pathways, biological processes and even cellular compartments disturbed by variables such as drug abuse. Nevertheless, the use of a large fraction of the archived specimens may be limited for determining global transcriptional changes due to problems with RNA quality. Postmortem time interval prior to tissue harvest is one of the main causes of poor RNA quality and presents a challenge to retrieve transcriptional information from a large number of human specimens, particularly the ones from drug users [1,2,3]. To address this challenge, we investigated whether changes in epigenetic marks such as enhancer histone modifications may be stable surrogates to indicate consequential information about gene transcription patterns, which would otherwise be lost due to poor RNA quality in postmortem specimens. This could enhance the investigation of global changes associated with HIV infection and neurological outcomes, as well as pathways involved in addiction, in a larger number of specimens.

Epigenetic marks associated with transcription include enhancer histone modifications such as H3K4me3 and H3K27Ac in proximal promoters, as well as the presence of RNA Polymerase I/II at transcription starting sites (TSS) [4,5]. These and other epigenetic mechanisms, either enabling or silencing transcription, are highly conserved between eukaryotic species [6,7,8,9]. We tested the effect of postmortem intervals on the stability of these epigenetic marks in mouse brain tissue and confirmed that the most stable modification over time was consistently found in human postmortem specimens with poor RNA quality. This information was used to assess signatures in prefrontal cortex specimens from HIV+ subjects that met the criteria of Methamphetamine (Meth) use disorder, or not.

We found that both histone modifications were better preserved than RNA Polymerase and that H3K4me3 was more abundant than H3K27Ac. H3K4me3 was consistently found in postmortem human specimens and allowed the identification of promoter and gene activity in inflammatory and viral response genes, as well as in genes involved in addiction. We conclude that the characterization of the epigenetic landscape can enable the retrieval of useful information from precious specimens with otherwise poor RNA quality, offering an opportunity to expand the analysis of specimens in tissue repositories, such as the National NeuroAIDS Tissue Consortium (NNTC). Importantly, this may help increase the number of analyzed specimens, enhancing sample size and statistical power, while improving models that incorporate common confounders and heterogeneities of the human population to genetic and post-translational patterns.

## 2. Materials and Methods

### 2.1. Mouse Postmortem Specimens and ChIP-qPCR

The studies were performed with the approval of the Institutional Animal Care and Use Committee (IACUC) of the San Diego Biomedical Research Institute (SDBRI). Eighteen 10 weeks old male C57Bl/6 mice were sacrificed using 5% isoflurane, and the bodies were maintained at room temperature for 0, 6, 24, 48, 72 and 96 h prior to dissection and brain harvesting. The brains were divided into two hemispheres, designated for RNA isolation and quality assessment performed as a service by the Microarray Core Facility at The Scripps Research Institute, La Jolla, as well as for chromatin isolation and cross linking. Mouse ChIP-qPCR was performed in isolated chromatin for the detection of H3K4me3, H3K27Ac and RNA Pol II with ChIP-grade antibodies. The reactions were performed in triplicate using 25 μg of mouse brain tissue chromatin and 3 μL of H3K4me3 antibody (Active Motif, Cat#39159, Carlsbad, CA, USA), 4 μg of H3K27Ac antibody (Active Motif, Cat#39133) or 4 μg of RNA Pol II antibody (Active Motif, Cat#91151). We performed qPCR using two positive control primers that worked well in similar assays (ACTB, GAPDH), as well as a negative control primer pair that amplifies a region in a gene desert on chromosome 6 (Untr6).

### 2.2. Human Postmortem Specimens and ChIP-qPCR

Frozen human prefrontal cortex brain specimens were provided by the National NeuroAIDS Tissue Consortium (NNTC) upon request and kept at −80 °C until experimentation. All experiments were performed with Institutional IBC and IRB approval (IRB-18-001-MCM). The prefrontal cortex specimens were selected among all male HIV+ cases (receiving ART and exhibiting criteria for global neuropsychological impairment [10,11,12,13], with available frozen and paraffin-embedded tissue. The specimens were divided in two groups based on meeting current stimulant abuse/dependence criteria at the last clinical evaluation and urine toxicology positive for methamphetamine (Meth− and Meth+, n = 3 case specimens/group). Three fragments of 1cm^2^ were separated from each frozen tissue specimen. Half of each fragment was used for RNA extraction and RNA quality assessment performed as a service by the Microarray Core Facility at The Scripps Research Institute, La Jolla. The remaining half was used for simultaneous chromatin isolation and cross-linking. ChIP reactions were performed using 30 μg of human HIV+ brain tissue chromatin and 4 μg of antibody against RNA Pol II, (cat#39097, Active Motif, Carlsbad, CA, USA), H3K27Ac (Cat#39085, Active Motif) and H3K4me3 (Cat#39160, Active Motif). Following that, qPCR was performed using two positive control primers (ACTB and GAPDH), as well as a negative control primer pairs that amplify regions in a gene desert on chromosome 12 (Untr12) for human specimens.

### 2.3. ChIP-qPCR Statistical Analysis

Statistical comparisons between postmortem intervals in ChiP-qPCR experiments in mouse and human preparations were performed using ANOVA, followed by Bonferroni’s posthoc test, in Prism 8.4.1 (Graphpad Software L.L.C., San Diego, CA, USA). Differences were considered statistically significant at *p* < 0.05.

### 2.4. RNA Extraction and Quality Assessment

Mouse and human tissue were homogenized with Kimble Kontes disposable pestles (Sigma-Aldrich, St. Louis, MO, USA), and RNA was extracted using RNAeasy Mini kit (Qiagen, Germantown, MD, USA). RNA quantification was determined by spectrophotometry using a NanoDrop ND-1000 spectrophotometer (NanoDrop Technologies Inc., Wilmington, DE, USA). The OD260/280 ratio was used to evaluate the purity of the nucleic acid samples, and the quality of the extracted total RNA was determined with the Agilent 2100 Bioanalyzer (Santa Clara, CA, USA), at the La Jolla Scripps Research DNA Array Core as a service.

### 2.5. Chromatin Preparation

Minced tissue was washed twice in ice cold PBS and treated with 1% formaldehyde (Sigma-Aldrich) for 12 min to crosslink the chromatin, as previously described [14,15]. The reaction was stopped with glycine added to a final concentration of 0.125 M. The pellets were lysed with lysis buffer containing molecular grade 85 mM KCl, 0.5% NP40 and 5 mM HEPES pH 8.0) supplemented with a protease inhibitor cocktail, incubated on ice for 15 min and centrifuged at 3500× *g* for 5 min to pellet the nuclei. The pellet was resuspended in nuclear lysis buffer (10 mM EDTA, 1% SDS, 50 mM Tris–HCl, pH 8.1) at a ratio 1:1 (*v*/*w*), incubated on ice for 10 min and stored at −80 °C until use. The pellets were sonicated, and DNA was sheared to an average length of 300–500 bp. The chromatin extraction was performed using Chromatin IP DNA Purification kit (Active Motif, Carlsbad, CA, USA). Genomic DNA (input) was prepared by treating aliquots of chromatin with RNase, proteinase K and heat (65 °C) to reverse cross-linking, followed by phenol and chloroform extractions and ethanol precipitation. Pellets were resuspended in 10 mM Tris, 1 mM EDTA, and the resulting DNA was quantified on a Nanodrop spectrophotometer (Thermo Fisher Scientific, Chicago, IL, USA). Results were used to calculate the total yield.

### 2.6. ChIP-Seq

ChIP was performed using the ChIP-IT High Sensitivity Kit (Active Motif). ChIP-seq reactions were carried out with 30 ug of chromatin and anti-H3K4me3 antibody. The ChIP DNA was processed into an Illumina ChIP-Seq library and sequenced to generate reads, which were aligned to the *H. sapiens* genome annotation (Hg38 assembly) and unique aligns (removed duplicates) were obtained. A signal map showing fragment densities along the genome was visualized in the Integrated Genome Browser (IGB) and MACS peak finding was used to identify peaks. Control data was derived from positive and negative control alignments. The 75-nt sequence reads generated by Illumina sequencing (using NextSeq 500) were mapped to the genome using the BWA algorithm with default settings. Alignment information for each read was stored in the BAM format. Only reads that passed Illumina’s purity filter, aligned with no more than 2 mismatches and mapped uniquely to the genome were used in the subsequent analysis. In addition, duplicate reads were removed. Since the 5′-ends of the aligned reads represent the end of ChIP/IP-fragments, the tags were extended in silico using Active Motif software at their 3’-ends to a length of 150–250 bp, depending on the average fragment length in the selected 200 bp library size. To identify the density of fragments along the genome, the genome was divided into 32-nt bins and the number of fragments in each bin was determined. This information in bigWig was visualized in UCSC genome browser. Intervals with tag enrichment were identified by chromosome number, start and end coordinates using MACS [16] and SICER [17], corresponding to significant enrichments in the ChIP/IP data compared to random Input. For normalization, the tag number of all samples was reduced by random sampling to the number of tags present in the smallest sample, using default settings. To compare peak metrics, overlapping Intervals were grouped into merged regions, defined by the start coordinate of the most upstream Interval and the end coordinate of the most downstream Interval. In locations where only one sample had an Interval, this defined the merged region. After defining the Intervals and Merged Regions, their genomic locations along with their proximities to gene annotations and other genomic features were determined and presented in .xls spreadsheets.

### 2.7. ChIP Analysis

Genes with H3K4me3 peaks found within 10 kb of start/end were compared in the ChIP samples. Overlap between the samples was high (70–90% in pairwise comparisons). Active region identification settings were −7500 to 2500 from promoter start, with 10,000 upstream and downstream-margin. The comparison of all samples averaged by group and the peak metrics obtained in Bioconductor ChiPpeakAnno (www.bioconductor.org, 2020) [18] were used in combination with a present/absent peak call information to find genomic regions with H3K4me3 occupancy patterns. Sequencing statistics can be seen in Table 1. The average fragment densities for 28,826 human genes (from TSS to termination site) and the corresponding promoters (from −1000 to +1000 nt relative to TSS) were determined. Peak values in gene bodies were corrected by length. For the analysis, the data files were normalized to the same number of unique alignments without duplicate reads. Intervals were determined using the SICER algorithm at a cutoff of FDR1E-10 and a Gap parameter of 600 bp (which merges peaks located within 600 bp of each other). Gene intervals (peaks) were determined using the SICER algorithm at a cutoff of FDR1E-10 and a Gap parameter of 600 bp (which merges peaks located within 600 bp of each other into a single “island”). The overlap between the samples was high (70–90% in pairwise comparisons). The tag densities/signal metrics relative to known gene annotations were examined and used in systems analysis.

### 2.8. Systems Analysis

Ratio and log 2-fold-change were calculated using tag densities/signal metrics relative to known gene annotations to group average signal values, with a cut off > and <1.5-fold, and *p* < 0.05 in pairwise comparisons performed in licensed JMP Pro 12 software (SAS Institute Inc., Cary, NC, USA). Enrichment, processes and pathway annotations were performed using z scores by licensed Ingenuity Pathway Analysis software (IPA, Qiagen), confirmed by DAVID Bioinformatics Resources 6.8 (https://david.ncifcrf.gov, 2020) and then visualized using local search features in GeneMANIA plugin [19,20,21] (www.genemania.org) in Cytoscape 3.7.0 platform [22] (www.cytoscape.org, 2020) with *Homo sapiens* sources from Reactome [23] (www.reactome.org, 2020) and BioGRID_ORGANISM [24,25,26] (https://thebiogrid.org, 2020). Transcription factor-DNA interactions derived from merged peak regions were modeled using licensed TRANSFAC v2021.1 [27,28,29] (GeneXplain GmbH, Wolfenbuttel, Germany) with JASPAR 2020 position frequency matrix [30] and input provided in .BED format, restricted to *H. sapiens* nervous system.

### 2.9. In Situ Hybridization for HIV (vRNA) Detection

Paraffin-embedded sections of prefrontal cortex (from the same subjects) were provided by the NNTC upon request. RNAScope 2.5 HD assay (Advanced Cell Diagnostics, ACD) was performed. Briefly, pre-treatment was performed with RNAscope 2.5 HD Detection Kit (RED) (Cat# 322360), RNAscope 2.5 Pretreat Reagents-H202 and Protease Plus (Cat#322330), RNAscope Target Retrieval (Cat#322000) and RNAscope Wash Buffer (Cat#310091), following manual instructions. Probe *set* V-HIV1-clade B-C3 (Cat#425531-C3) targeting different segments within the *gag-pol* region, as described [31,32,33]. Images were captured using the light feature of a Zeiss AXIO Observer.Z1 (Carl Zeiss AG, Oberkochen, Germany).

## 3. Results

Peaks of enrichment in ChIP-seq data may suggest stable and predictive standards associated with transcription, especially the ones in promoter regions [5]. We selected 3 epigenetic marks associated with active transcriptional patterns: H3K27Ac, H3K4me3 and RNA Polymerase II (Pol). The stability of different epigenetic marks was tested in postmortem mouse specimens. Preservation of the most stable was confirmed in human postmortem brain specimens with low RNA quality and used to compare prefrontal cortex signatures of Meth use disorder in the context of HIV infection.

### 3.1. Stability of Enhancer Epigenetic Marks in Mouse Postmortem Brains Overtime

The stability of H3K27Ac, H3K4me3 and RNA Pol was tested in mouse brains, at different postmortem intervals, from 0 to 96 h time points. For that, C57Bl/6 mice were sacrificed and left at room temperature in groups of 3, for 6 h, 24, 48, 72 and 96 h before dissection and brain harvesting. Figure 1 shows representative RNA electrophoresis indicating bands and increasing fragmentation over higher postmortem time intervals. The 2 histogram peaks in *x*-axis designate the 18 s and 28 s ribosomal RNA, respectively. The *y*-axis indicates fragment units (FU) derived from band density in relation to standards. Notice that fragment units decrease by about 50% at 24 h compared to 0 and 6 h, although 18s and 28s peaks in histograms may retain aspects of integrity. At 0–6 h, RIN was 8.8 ± 0.5. From 24 h on, RIN was consistently <7.2, reaching 2.5 ± 0.3 at 96 h. RIN < 7.5 is considered unfit for RNA studies, as it signifies a significant fragmentation, indicating poor quality that results in inaccurate transcriptional data.

We used these mouse brain specimens to test the interaction between functional genomic marks in mouse brains and time in different postmortem intervals. For that, we used ChIP-qPCR to test H3K4me3, H3K27Ac and RNA Pol II onto promoters of two primer-amplified positive control genomic regions, corresponding to the housekeeping genes beta actin (ACTB) and glyceraldehyde-3-phosphate dehydrogenase (GAPDH), which are permanent and constitutively transcribed with a fixed number of occupancy units across samples, as well as a negative control primer pair that amplifies a region in a gene desert on chromosome 6 (Untr6), as recommended by standard quality control protocols [34]. Figure 2 shows the relative increment of H3K4me3, H3K27Ac and RNA Pol II binding events in the amplified ACTB and GAPDH genes compared to Untr6 negative control genomic region. RNA Pol became undetectable faster than H3K27Ac and H3K4me3, indicating less stability. Both histone modifications had a decrease at 24 h, were similar between 24 and 72 h and suffered a drastic loss at 96 h. H3K4me3 was the modification with stronger signal over input, as shown in the increase of 200-fold above input at the GAPDH gene. This indicates that H3K4me3 may have a better detection stability up to 72 h of postmortem interval.

### 3.2. Stability of Functional Genomic Marks in Human Postmortem Prefrontal Cortex Specimens with RIN < 7.2

We hypothesized that H3K4me3 would be consistently detected in human specimens with poor RNA quality not considered adequate for transcriptional studies. We tested the same marks, H3K4me3, H3K27Ac and RNA Pol, in human postmortem prefrontal cortex specimens, which had a poor RNA quality (RIN < 6.2) and were not considered adequate for transcriptional studies. The specimens were obtained from the NNTC and were from ART-treated male subjects from the UCLA cohort between 2003 and 2011, following inclusion criteria described in methods. Table 2 shows subject characteristics that were relevant for grouping.

Detection of viral RNA in matching paraffin embedded tissue sections by RNAscope was detectable, although not consistently in all the specimens (Figure 3). Viral DNA was not identified.

Figure 4 shows that a stronger increase over input was observed in both histone modifications compared to RNA Pol, with better results for H3K4me3. Like in mice, H3K4me3 had a 200-fold increase in signal over input at the GAPDH gene, indicating cross-species consistency.

### 3.3. Quality Controls in Genome-Wide Peak Data Confirms H3K4me3 Stability in Postmortem Human Specimens

Given the relative stability of H3K4me3 in mouse and human specimens, this genomic unit was then selected for comparing two groups of HIV+ brain specimens represented by 6 cases with HIV-associated neurological diagnosis, divided in 2 subgroups, being 3 cases with no history of substance use and 3 cases that fit Meth-use criteria, based on self-reported recent and life-time use. Table 1 shows the patient characteristics associated with the specimens used in signature comparisons. All unidentified subjects were 20–40-year-old males. Viral load at CD4 nadir, ART information and observations from available neuropsychological (NP) testing are shown in Table 1. Prefrontal cortex specimens from these subjects had RIN values below 7.2. These were used to test the hypothesis that epigenetic marks are sufficiently stable to provide information from specimens that would otherwise have a limited value for the study of global changes.

The results in Figure 4A suggest that H3K4me3 signal is stronger than H3K27Ac, although both are stable in postmortem specimens up to 72 h postmortem. Figure 4B shows H3K4me3 peaks in representative specimens, at the GAPDH gene.

The quality of genome-wide H3K4me3 signal across samples was tested using different methods (Figure 5). Active regions were merged within and between groups, revealing that specimens in both groups were stable regarding location of peaks relative to genomic coordinates and annotations mapped to the Human Hg38 assembly and visualized in pie charts (Figure 5A). Tag distribution occurred predominantly in proximal promoters and introns, indicating that genomic activity was likely to be associated with transcriptional events, but also in regulatory regions, as opposed to Hg38 random controls in introns and non-coding regions (Figure 5A). Differences in peak intensity have identified mutual as well as exclusive signatures, indicating that this strategy can distinguish between groups (Figure 5B). For instance, 8525 active genomic regions were identified in HIV+Meth- specimens, which were not present in Meth users. Conversely, HIV+Meth+ specimens presented 7857 exclusive active regions. The majority of the active genomic regions were shared by specimens in both groups. The number of active regions per chromosome was also similar across cases and between groups (Figure 5C), with large activity in chromosomes 1 and 19 and comparatively low activity in chromosomes 18, 21 and Y.

Figure 6 shows the H3K4me3 peak distribution patterns in both groups of specimens, both in average plots and heatmaps from tag distributions across target regions such as gene bodies for all target regions (Figure 6A,D), mapping merged regions for all peak regions (Figure 6B,E), and transcription start site (TSS, at 0 bp) proximal promoters (Figure 6C,F). The heatmaps (values in z-axis/color, regions in *y*-axis) are used for visualization of read densities or H3K4me3 tag distributions (using bigWIG metrics), which are mapped across target regions. Peaks mapped to gene bodies located between −2 kb and +2 kb, excluding distal downstream and upstream peak signals (Figure 6D). Merged Regions correspond to all peak regions from −5 kb to +5 kb to include distal promoter and regulatory regions (Figure 6E). Peaks in proximal Promoters were located near the TSS, between −1.5 kb and +1.5 kb (Figure 6F). These heat maps show H3K4me3 distribution patterns that characterize different genes and active genomic regions. The heatmaps were clustered into 5 groups based on where tags were located within the gene sequence, using default settings. The clusters C1–C5 differed between gene body, merged and promoter heatmaps, due to the cut-off distance from the TSS used to build each one of them. These patterns are not expected to differ at least drastically between groups, in spite of the differences in Meth use, which might be detected rather by a granular analysis of peak intensities. Thus, the comparison between HIV+Meth- and HIV+Meth+ in Figure 6 suggests that specimens in both groups followed similar and normal pattern distributions with parallel clusters, indicating no aberrant antibody binding and signal above pooled input controls, adding up to build confidence on data quality. The comparison can be also established and visualized as average plots (Figure 6A–C), which average the values for all target regions in heatmaps into histograms and indicate tag distribution in reference to the TSS in *x*-axis. Consistency across specimens was further confirmed from a pairwise comparison of the averaged tag numbers for merged regions in the HIV+Meth- and HIV+Meth+ groups; a scatter plot was produced with a Pearson correlation coefficient = 0.958 and a slope = 0.977.

### 3.4. H3K4me3 Can Provide Insights into Biological Processes and Pathway Usage in Specimens with Limited Value in Global Transcriptome Studies

We used H3K4me3 to test our ability to use epigenomics as an alternative to transcriptome strategies and increase power in archived human tissues. Peak values were averaged, and fold change between HIV+Meth- and HIV+Meth+ groups was calculated. The strong signal of H3K4me3 and differences between groups were detectable, as exemplified in Figure 7, showing a representative randomly selected genomic region associated with the AATK (Apoptosis-associated tyrosine kinase) gene sequence in chromosome 17. A representative specimen from the HIV+Meth+ group showed a higher H3K4me3 peak in the proximal promoter region compared to a representative HIV+Meth- specimen, indicating that promoter activity is higher for this gene in the context of Meth, suggesting it is more likely to be transcribed in drug users, while activity within the intron did not differ between the specimens.

The application of a systems biology approach via Genemania and IPA has suggested that this strategy is able to provide important information on functional processes and pathway usage, while mapping effects of infection and co-morbidities on functional units of the genome. Overall, we found 6062 active intervals located upstream the gene sequence, of which 2210 were within proximal promoter regions (−1 kb–0) (Figure 6F, C2). Of these, 1540 showed a high degree of connectivity based on pathway and physical interactions, a large fraction of them with detectable changes in peak intensity between specimens of the two groups (Figure 8). An approach limited to genes containing H3Kme3 in proximal promoters produced significant annotations in OMIM_DISEASE that include somatic cancer (*p* = 0.0003) and susceptibility to obesity (*p* = 0.006). Overrepresented functional annotations are within phosphoproteins (*p* = 5.3 × 10^−103^), alternative splicing (*p* = 5.8 × 10^−31^) and ubiquitin conjugation (*p* = 4.7 × 10^−28^). Biological processes (GOTERM) include positive regulation of GTPase activity (*p* = 6.8 × 10^−10^), Regulation of transcription (*p* = 3.2 × 10^−9^), Wnt signaling pathway (*p* = 1.8 × 10^−8^), Cell migration (*p* = 2.4 × 10^−7^), MAPK cascade (*p* = 1.7 × 10^−7^), neuron projection development (*p* = 3.0 × 10^−6^), positive regulation of apoptosis (*p* = 4.3 × 10^−6^), oxidative response (*p* = 6.8 × 10^−5^) and glial cell differentiation (*p* =6.6 × 10^−4^). Pathway annotations (KEGG) for genes with proximal promoter epigenetic activity are Pathways in cancer (*p* = 2.5 × 10^−16^), several pathways involved in neurotransmitter signaling and neurological pathogenesis, such as cholinergic synapse (*p* = 1.5 × 10^−7^), oxytocin signaling (*p* = 6.8 × 10^−7^), glutamatergic synapse (*p* = 9.4 × 10^−7^), dopaminergic synapse (*p* = 3.9 × 10^−6^), cAMP signaling (*p* = 5.6 × 10^−5^), amphetamine addiction (*p* = 1.2 × 10^−5^), serotoninergic synapse (*p* = 9.4 × 10^−4^), morphine addiction (2.7 × 10^−3^) and endocannabinoid signaling (*p* = 2.5 × 10^−3^). Inflammatory and chemokine pathways were also identified (*p* = 1.4 × 10^−5^). This indicates that epigenetic strategies like this can distinguish disruptions in neurotransmitter signaling.

We have identified 2210 genes with active proximal promoter, with both increased (orange shades) and decreased (blue shades) promoter activity associated with Meth use and white nodes corresponding to no change (Figure 8). In Figure 8, it is important to notice the high degree of connectivity between the genes that have H3K4me3 activity, indicating that genomic activity is an orchestrated phenomenon. Nevertheless, many genes with significant proximal promoter activity did not show functional or physical interactions with each other and remained independent. A complete list of these genes and fold changes between groups is available in Appendix A.

The genes in Figure 8 were further filtered based on a significant +/− 1.5-fold change cut off in a comparison between HIV+Meth- and HIV+Meth+. Tight clustering criteria produced two high score nodes, calculated in Cytoscape default network topology statistics settings, based on the number of neighboring nodes and the number of connected pairs between all neighbors. The first, a 5.2 score node, indicates a cluster where each gene represented by node circles have an average of 5.2 related neighboring genes and with interdependent functional domains based on genomic (pink edge connectors), physical (red) and pathway interactions (green connectors) (Figure 9A). The genomic regions annotated to genes in this cluster exhibited both increased (orange nodes) and decreased (blue nodes) promoter activity, or no change (white nodes) in correlation with Meth use (Figure 9A). In order to further visualize the potential implications of those changes and relationships between these genes, we performed a functional annotation cluster analysis (Figure 9B), which identified biological processes as well as transcription regulators (Figure 9B). For instance, increased promoter activity associated with Meth included genes annotated to acute inflammation, macrophage infiltration, cell movement of antigen presenting cells as well as necrosis, apoptosis, and cell death of muscle cells. These changes were predicted as being orchestrated by IFNG, which showed a significant 2.3-fold increase in its promoter genomic activity (*p* < 0.0001). Other IFN pathway components such as IRF1 (1.8-fold, *p* < 0.0001) or transcriptional regulators JUN (1.8-fold, *p* < 0.001) and SOX2 (1.8-fold, *p* < 0.001) were also identified as key components that showed significant changes in the context of Meth (Figure 9B). Overall, this suggests that the inflammatory process is enhanced in HIV+Meth users, in an IFN-dependent manner. Interestingly, genes with decreased promoter activity (shades of blue) shared connections with genes with increased promoter activity, based on physical protein–protein and genetic interactions, as well as shared domains or pathway, as shown in Figure 9A, indicating potential regulators. Neurological pathways involved memory and cognition, as well as CREB signaling in neurons (Figure 9B) were identified. In Figure 9B, edges or connectors are directional.

A second significant high-score node (4.8 score) identified among genes with proximal promoter activity and significant ><1.5-fold changes was characterized by dopaminergic synapses signatures (Figure 9C), also including genes with both increased and decreased activity. In this node, decreased activity in the Tyrosine Hydroxylase (TH) and PAX7 promoters suggests a loss of dopaminergic function and neural progenitors in Meth users in the context of HIV [35]. In both high score nodes (Figure 9A,C), the color-coded line edges show the interaction criteria between the genes of interest, in these cases based predominantly in genetic and pathway interactions and indicating a coordinated process.

The unbiased analysis of all genes with HeK4me3 active regions (Figure 6E), regardless of position (in gene, downstream, upstream proximal or upstream distal, C1–C5), using a 1.5 cut off and 95% confidence level in detectable changes caused by Meth in the context of HIV, has led to the identification of several gene clusters and pathways annotated to addiction (Figure 10) and neuronal synapses (Figure 11), expressing a high degree of redundancy, which is evidenced by merging features. The systems analysis indicates differential activity in genes annotated to nicotine (Figure 10A), morphine (Figure 10B), amphetamine (Figure 10C) and cocaine (Figure 10D) addiction. These pathways have a high degree of overlapping interactions that could be merged into a single network (Figure 10E).

Clusters annotated to synaptic pathways have also been able to distinguish HIV+Meth+ from HIV+Meth-, using the same approach including all merged HeK4me3 active regions (Figure 6E). For instance, high score gene clusters were annotated to glutamatergic (Figure 11A), dopaminergic (Figure 11B), cholinergic (Figure 11C), serotoninergic (Figure 11D), as well as GABAergic synapses (Figure 11E). Gene nodes were connected by protein–protein and genetic interactions, as well as pathway. Other high score pathways were annotated to neuronal functions, such as long-term potentiation (Figure 12A) and neuroactive ligand-receptor interactions (Figure 12B). The overlap between synaptic and neuroactive pathways was revealed by merging feature properties (Figure 12C). The large number of neural synapses genes with blue node color suggests that H3K4me3 is decreased in the context of Meth.

Interestingly, like with addiction pathways, neurological synapses genes showed a lower number of H3K4me3 active peak regions in the prefrontal cortex from HIV+Meth users compared to non-Meth users, suggesting differences in enhancer chromatin remodeling events, while epigenomic alterations such as the one we are measuring have the capacity to affect the behavior of the identified gene networks. However, epigenomic factors do it so in cooperation with transcription factors and the respective binding motifs made available by such modifications. ChIP data input in BED format into TRANSFAC generated a functional analysis report where prefrontal cortex was the most significantly over-represented attribute (*p* = 2.29 × 10^−26^). Cell type attributes characterized by the over-representation patterns included neurons (*p* = 1.86 × 10^−12^), followed by T cells (*p* = 3.93 × 10^−8^), while innate immune cells (macrophages/microglia) were significantly under-represented (*p* = 3.77 × 10^−6^), suggesting potential dysfunction in that compartment regardless of drug use patterns.

To further help in the prediction of changes in the behavior of gene networks, and the effects of addiction, we modeled transcription factor usage to estimate the contribution of transcription factors affected and recruited by H3K4me3 in gene intervals, added 200bp flanking regions. For that, we used TRANSFAC Match algorithm, which determined the frequency of transcription factor binding motifs in H3K4me3 intervals aligned to the GRCh38 human genome assembly from the Genome Reference Consortium (GCA_000001405.15 GCF_000001405.26). Table 3 shows the 10 most frequent binding motifs, significantly represented in active genomic regions in both HIV+Meth- and Meth+ group specimens.

## 4. Discussion

Postmortem specimens from humans are a valuable resource for understanding molecular dynamics taking place in a diversity of conditions. Even more critical is the fact that large association studies benefit from the integration with transcriptional profiles. However, RNA quality has been a limitation, encountered by our group and others [2,36,37]. It has been shown that small differences in RNA integrity affect gene expression quantification by introducing a moderate and pervasive bias in expression level estimates [38]. Postmortem interval is a major factor affecting RNA quality, along with years of storage [36,39]. This is an important issue particularly in specimens from substance user populations, with very diverse characteristics of interest [40]. Our results suggest that predictions on the behavior of gene networks can be derived from specimens that have a limited use in transcriptional profiling studies, by comparing epigenetic marker patterns that are preserved over longer postmortem intervals [41].

Epigenetic modifications, enhancers and regulators of transcriptional activity are likely protected by protein-based mechanisms that prevent physical damage to the genome and evolutionarily tailored to maintain the stability of gene expression patterns [42]. We tested the postmortem availability and preservation of two enhancer histone modifications, H3H4me3 and H3K27Ac, as well as RNAPol II at the TSS, all regarded as strong correlates of gene expression activity, using ChIP in mouse and human brain specimens [43,44,45].

In our studies in mice with controlled postmortem intervals, we found that H3K4me3 was more abundant than other active chromatin modifications, followed by H3K27Ac, and by RNAPol binding. Although this may reflect a difference of stability [41] developed for the preservation of genomic information, other factors might explain a significant relative difference in the abundance of these marks. For instance, the abundance of H3K4me3 could be associated to the number of histone modifications required to produce changes in the chromatin architecture and prime RNAPol recruitment as a single event. It may be also possible that not every histone modification will result in productive remodeling, which may depend on additional factors. Nevertheless, differences in stability are suggested by the negative slope. For instance, the slope of RNAPol compared to histone modifications indicates a faster decay or loss. Nevertheless, at 96 h postmortem, significant loss in all measures indicated that 72 h was the maximum postmortem interval for retrieving reliable data. H3K4me3 was more abundant than H3K27Ac and was detectable in postmortem tissues with larger intervals up to 72 h. The H3K4me3 epigenetic mark was also consistently found in human brain specimens with low RNA quality and used to examine frozen prefrontal cortex specimens preserved from HIV+ subjects.

A higher activity based on the number of H3K4me3 active sites was found in chromosome 1, which is the largest human chromosome and reported involvement in heritable brain disorders [46]. Similarly, the highly active chromosome 19 contains genes such as APOE and Notch 3, whose polymorphisms have implications in, respectively, Alzheimer’s disease (AD) [47,48] and stroke and dementia associated with cerebral autosomal dominant arteriopathy with subcortical infarcts and leukoencephalopathy [49,50]. HIV neurocognitive disorders share biomarkers and have similarities with AD [51]. Nevertheless, Meth had no effect on APOE gene activity. The similarity regarding mean gene activities within different chromosomes and across subjects indicates that potential differences between the groups are likely not due to disrupted enhancing epigenetic mechanisms, and they can be used for comparisons. This is also evidenced by similar H3K4me3, H3K27Ac and RNAPol tag ratios in housekeeping genes between groups.

The analysis of H3K4me3 peak distribution was used predict signatures of Meth use disorder in the context of HIV. H3K4me3 promoter and gene activities are associated with transcriptional events in several systems [43,52,53]. Systems biology approaches were performed using activity that is exclusively in proximal promoters as well as including all gene regions. The majority of the in-promoter activity occurred in genes clustered in a large interactive network and annotated to inflammatory and viral responses, as well as to the dopaminergic system. Meth use was associated with enrichment of genomic activity in process-relevant genes such as IFNG and AATK, as well as loss of TH promoter activity [54], which is potentially associated with dopaminergic deficits. This finding suggests that Meth use increases genomic activities most likely to become transcriptional events that increase inflammation and apoptosis, with implications to neurodegeneration, dopaminergic deficits and decreased cognitive and executive functions, confirming previous observations from animal models of neuroHIV and from humans [55,56,57,58,59,60,61]. This indicates that epigenetic signatures are able to retrieve biologically meaningful data replicated by other systems and have the potential to provide incremental knowledge in association studies. In addition, this also suggests that biological processes associated with pathogenesis in the context of HIV and in drug users may have an epigenetic basis.

The majority of the promoter and gene activity evidenced by H3K4me3 peaks, was suppressed in HIV+ Meth users, compared to HIV+ non-Meth users. These results also parallel previous transcriptional findings in a mouse model of HIV and Meth abuse, where Meth caused a significant suppression of gene transcription [57,62], particularly regarding the dopaminergic system [57,62,63]. This was also evident when the analysis included H3K4me3 peaks in all different gene regions, including downstream, upstream distal and upstream proximal (promoter region), as well as in exons, addiction and neuronal synapses gene networks were specifically identified, suggesting that genomic activity measures like the one we performed here can help predict changes associated with addiction and synaptic disorders. For instance, we distinguished pathways involved in drugs of abuse, in addition to amphetamines, with a granular precision based on various degrees of interaction. Among the pathways showing changes in gene activity in correlation with Meth use, we have identified nicotine, cocaine and opioids. We cautiously interpret this finding given the limited knowledge about the subjects’ history. However, polysubstance use is indeed prevalent among Meth abusers [64] and may be a factor. If this is the case, our strategy has the power to distinguish addictive disorders with a high degree of granularity. However, it is also important to acknowledge that molecular mechanisms involved in addiction have commonalities between different drugs [65,66,67]. This is evident in the overlap between addiction pathways, with common elements in reward circuitry and neurotransmitters.

The gene networks associated with individual synaptic functions exhibited a common behavior, characterized by an overall decrease in genomic activity in Meth users in the context of HIV. This could result from mechanisms that favor and enhance the inflammatory process, which were identified here, leading to neuronal loss, including in the dopaminergic system, a hallmark of Meth and other substance use disorders [68,69]. Importantly, the patterns associated with Meth use strongly suggest a decrease in synaptic strength. This weakening was detectable in both inhibitory and excitatory γ-aminobutyric acid (GABA) and glutamatergic synapses projected from other brain regions to the prefrontal cortex, which was examined here. Along with dopaminergic projections into the prefrontal cortex, neuroactive ligand-receptor interactions play an important role in cognition and learning [70], temporal organization of behavior [71,72,73,74], as well as executive functioning that form the basis of planning, reasoning and thinking. In this area, a balance in neurotransmitters and neural networks is critical for the stability of the neuronal activation in that brain region [75,76,77]. Similar changes in circuitry have been previously described in correlation with Meth and other psychostimulants [78]. Moreover, the dopaminergic system is independently affected by HIV, with a described impact on addiction behaviors [79,80]. While our work suggests an epigenetic basis for neuronal circuitry disorders, it also shows such disorders can be distinguished in the brain of Meth users in the context of HIV based on epigenomic activity.

In addition to the prediction of gene network behaviors, patterns of H3K4me4 peak activity can be used to predict transcription factor usage patterns, based on the occurrence of binding motif sequences within and near active intervals. We have performed this analysis and found that Sp1 was the most frequent binding motif. It has been shown that Sp1 makes a complex with NFkB to enhance HIV replication in vitro, in a process that is facilitated by Meth [81]. The second most abundant binding motif was p53, which has been described as a critical transcription factor in long term neurotoxic effects of Meth, particularly in dopaminergic neurons [82]. ETS2, which was another enriched binding motif, has been described as a transcriptional repressor of IL2 as well as of HIV, as it binds to the HIV-LTR-RATS element [83,84]. Thus, transcription factor binding predictions suggest potential important Meth effects on HIV replication and neurotoxicity.

This study has limitations, including small sample size and the lack of HIV-negative specimens. Our epigenetic findings can inform processes occurring in the human brain but must be integrated with transcriptome data in specimens and models with higher RNA quality stability for validation. Nevertheless, since a large fraction of the H3K4me3 genomic activity occurs in proximal promoters followed by introns, it is likely to result into and correspond to transcriptional events.

We demonstrate that important insights can be gained from epigenetic approaches and opens to the opportunity of investigations with larger sample sizes. We have shown the potential of using enhancer epigenetic marks for making predictions on the behavior of gene networks and for recovering important information from a significant number of specimens with limited use due to postmortem or storage intervals.

## Figures and Tables

**Figure 1 viruses-13-00544-f001:**
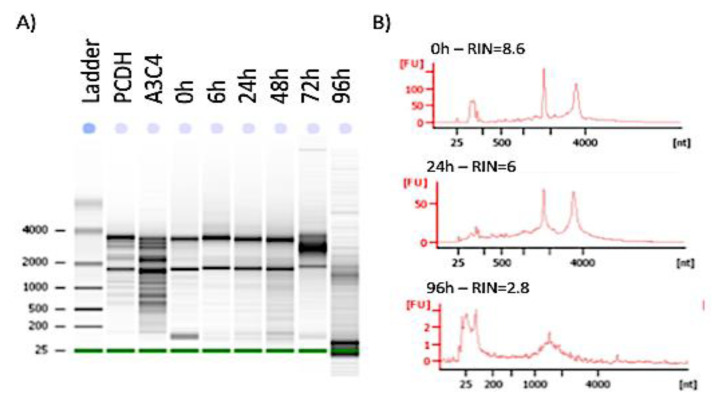
RNA stability in mouse brains. C57Bl/6 mice were sacrificed and left at room temperature for 0, 6, 24, 48, 72 or 96 h before dissection (n = 3/interval). PCDH and A3C4 are Agilent RNA integrity reference standards. Their brains were harvested, and half was snap frozen for RNA extraction. The RNA quality was tested by (**A**) gel electrophoresis, in an Agilent Bioanalyzer 2100. (**B**) RNA integrity numbers (RIN) were derived from 18 s and 28 s fragment units (FU) in experimental specimens and reference standards, as shown in 0, 24 and 96 h representative histograms. RIN < 7.5 is considered unfit for RNA studies, as it indicates a significant fragmentation.

**Figure 2 viruses-13-00544-f002:**
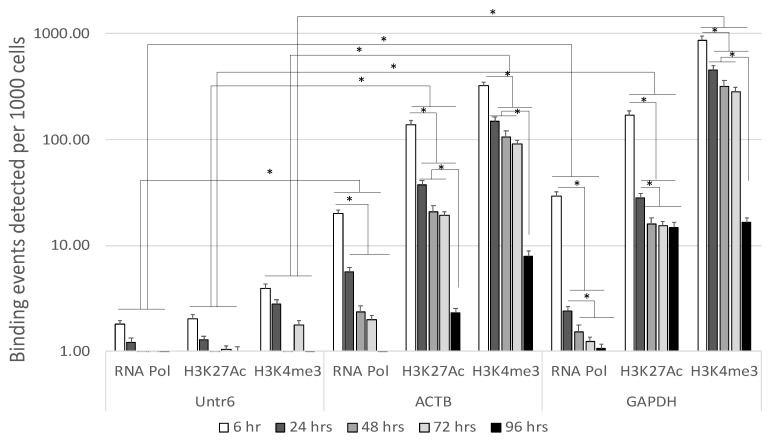
Postmortem stability of genomic epigenetic marks positively associated with transcription. Mice were sacrificed and maintained at room temperature for 6, 24, 48, 72 and 96 h prior to the dissection and brain harvesting. ChiP-qPCR was performed in isolated chromatin for the detection of H3K4me3, H3K27Ac and RNA Pol II with ChIP-grade antibodies. The reactions were performed in triplicate using 25 μg of mouse brain tissue chromatin and 3 μL of H3K4me3 antibody (Active Motif, Cat#39159), 4 μg of H3K27Ac antibody (Active Motif, Cat#39133) or 4 μg of RNA Pol II antibody (Active Motif, Cat#91151). We performed qPCR using two positive control primers that worked well in similar assays (ACTB, GAPDH), as well as a negative control primer pair that amplifies a region in a gene desert on chromosome 6 (Untr6). * *p* < 0.05 in comparisons indicated by lines.

**Figure 3 viruses-13-00544-f003:**
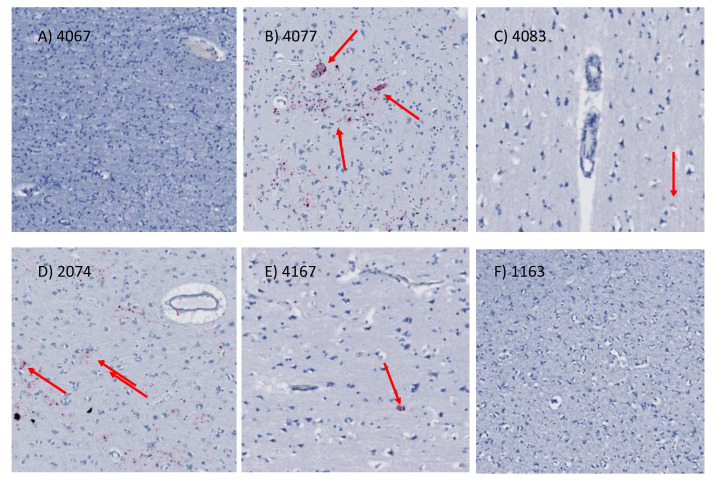
RNA scope in situ hybridization showing HIV-infected cells in prefrontal cortex. HIV vRNA probes were used to detect active virus. Representative sections from the prefrontal cortex are shown for specimens (**A**) 4067, (**B**) 4077, (**C**) 4083, (**D**) 2074, (**E**) 4167 and (**F**) 1163. Red arrows show examples of positive staining patterns in pink, which correspond to expression of HIV vRNA in the brain parenchyma. Magnification 60X.

**Figure 4 viruses-13-00544-f004:**
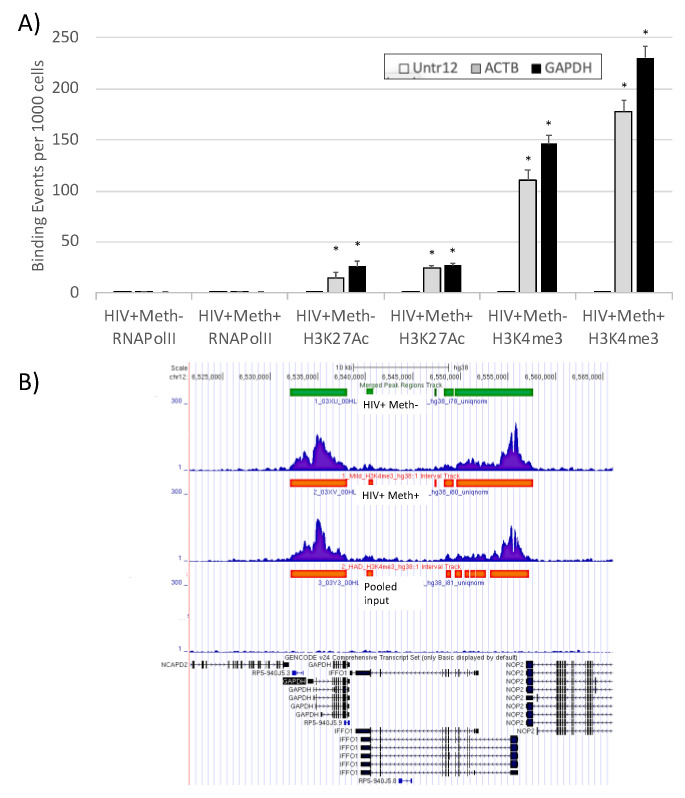
Comparison of detectable epigenetic marks in HIV+ prefrontal cortex specimens. Using ChIP-qPCR, we tested our ability to detected genomic marks in postmortem human prefrontal cortex tissue with low RNA quality. (**A**) RNAPol, H3K27Ac and H3K4me3 were detected in 3 amplified genomic regions: A desert sequence in the human chromosome 12 was used as negative control (white bars), ACTB (gray bars) and GAPDH (black bars) promoters. * *p* < 0.05 compared to Untr12 negative control. (**B**) H3K4me3 peaks in the GAPDH genomic sequence in representative postmortem specimens from both groups. * *p* < 0.05 in comparisons indicated by lines.

**Figure 5 viruses-13-00544-f005:**
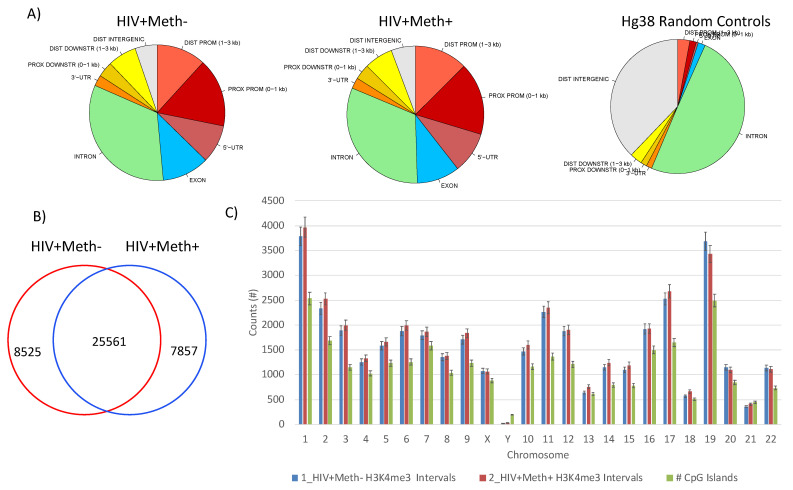
Mapping of HeK4me3 active genomic regions in human postmortem brain specimens. (**A**) Pie charts showing the distribution of H3K4me3 in gene sequences in both groups, with genome wide frequency of modifications within introns, exons, 5′-UTR, proximal (0–1 kb) and distal (1–3 kb) promoters, distal intergenic promoter regions, proximal (0–1 kb) and distal (1–3 kb) downstream regulatory regions and 3′-UTR. As a control, randomly located peaks were also ran against the same genomic features database using the entire interval region aligned to the Human Hg38 genome assembly. (**B**) Venn’s diagram showing genome wide exclusive and mutual merged active regions in HIV+Meth- and HIV+Meth+ cases (n = 3/group). (**C**) Number of HeK4me3 active peak intervals and CpG islands per chromosome in averaged HIV+Meth- and HIV+ Meth+ cases.

**Figure 6 viruses-13-00544-f006:**
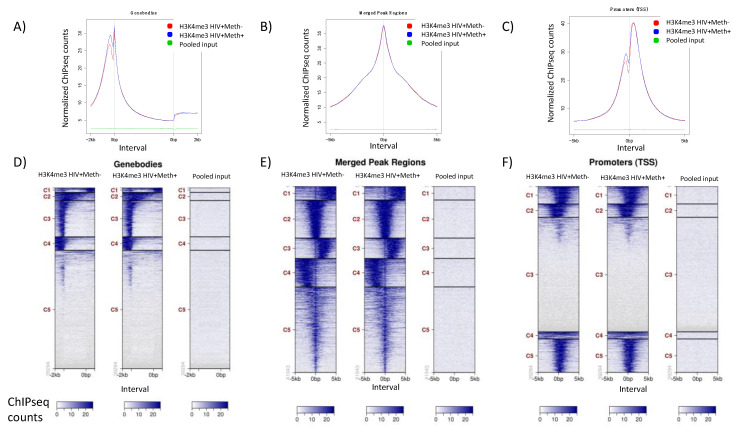
Average plots and heatmaps. Average plots and heatmaps were produced from aggregated peak scores indicating tag distributions (using bigWIG metrics) across target regions identified as (**A**,**D**) gene bodies (+/−2 kb), (**B**,**E**) Merged Regions (+/−5 kb) and (**C**,**F**) proximal to transcription start site (TSS) promoter regions (+/−1.5 kb). (**A**–**C**) Average plots correspond to histograms of the average distribution of all regions. (**D**–**F**) Values in heatmaps are represented in z-axis/color and regions in *y*-axis, using a 5-cluster default, indicated by C1–C5 in *y*-axis for each region of interest. The gradient blue-to-white color indicates high-to-low count in the corresponding interval region.

**Figure 7 viruses-13-00544-f007:**
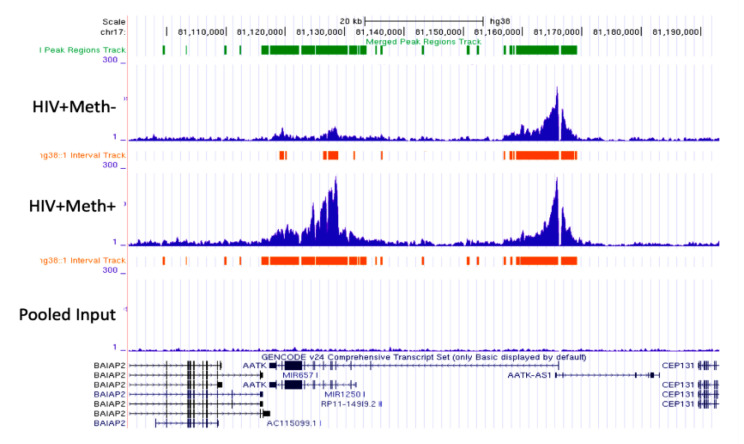
Example of detectable H3K4me3 tag differences between HIV+Meth- and HIV+Meth+ specimens. AATK Gene (apoptosis-associated tyrosine kinase), with similar marks in the gene regulatory region, but enhanced at the promoter in a HIV+Meth+ representative specimen. Visualized using Integrated Genome Browser.

**Figure 8 viruses-13-00544-f008:**
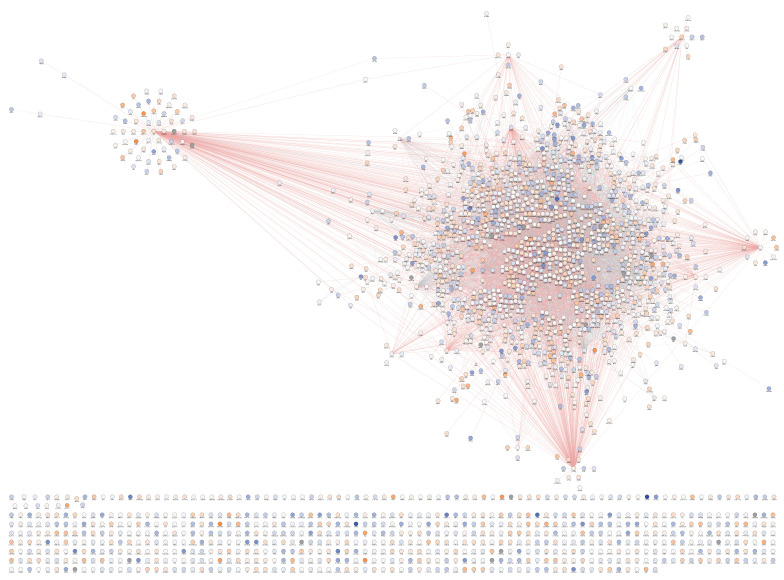
Clustering of genes with active proximal promoter merged regions. A total of 6062 active intervals were located upstream the gene sequence, of which 2210 were within proximal promoter regions (−1 kb–0) indicating pro-transcriptional activity. Connectivity based on pathway (blue connectors) and physical interactions (red connectors) produced a cluster of 1540 genes. Changes in peak intensity in HIV+Meth+ compared to HIV+Meth- are shown by color (blue–decrease, orange–increase, white–no change).

**Figure 9 viruses-13-00544-f009:**
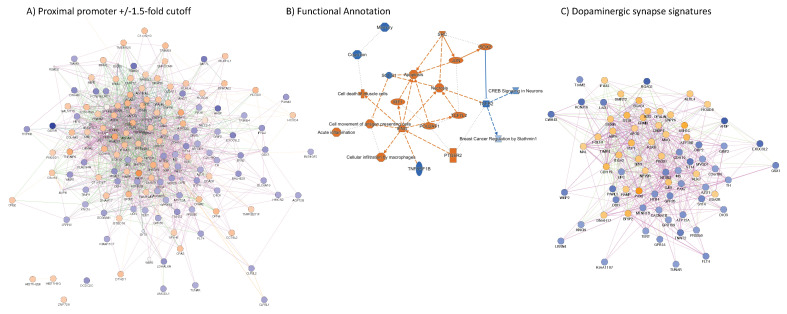
Network of genes with significant changes in H3K4me3 dynamics within the proximal promoter in HIV+Meth+ postmortem prefrontal cortex specimens compared to HIV+Meth-. (**A**) A weighted gene network from in-promoter enrichment patterns filtered to +/−1.5-fold differences, where circles in shades of blue indicate decrease, and shades of orange indicate enrichment, with relationships based on genetic interactions (pink edges), physical interactions (red edges) and pathway (green edges). (**B**) Functional Annotation derived from IPA related to genes with in-promoter modifications and prediction of transcriptional regulators. Edges indicate directional molecular and pathway (orange) and biological processes (blue) relationships. (**C**) Subnetwork containing dopaminergic synapse signatures showing in-promoter modifications, identified by grouped by genetic interactions (pink edges), physical interactions (red edges) and pathway (green edges) (KEGG, *p* = 3.9 × 10^−6^). Node sizes represent connection scores. Blue colors indicate decrease and orange indicates increase in H3K4me3 peak signals in averaged HIV+Meth+ compared to HIV+Meth- prefrontal cortex specimens.

**Figure 10 viruses-13-00544-f010:**
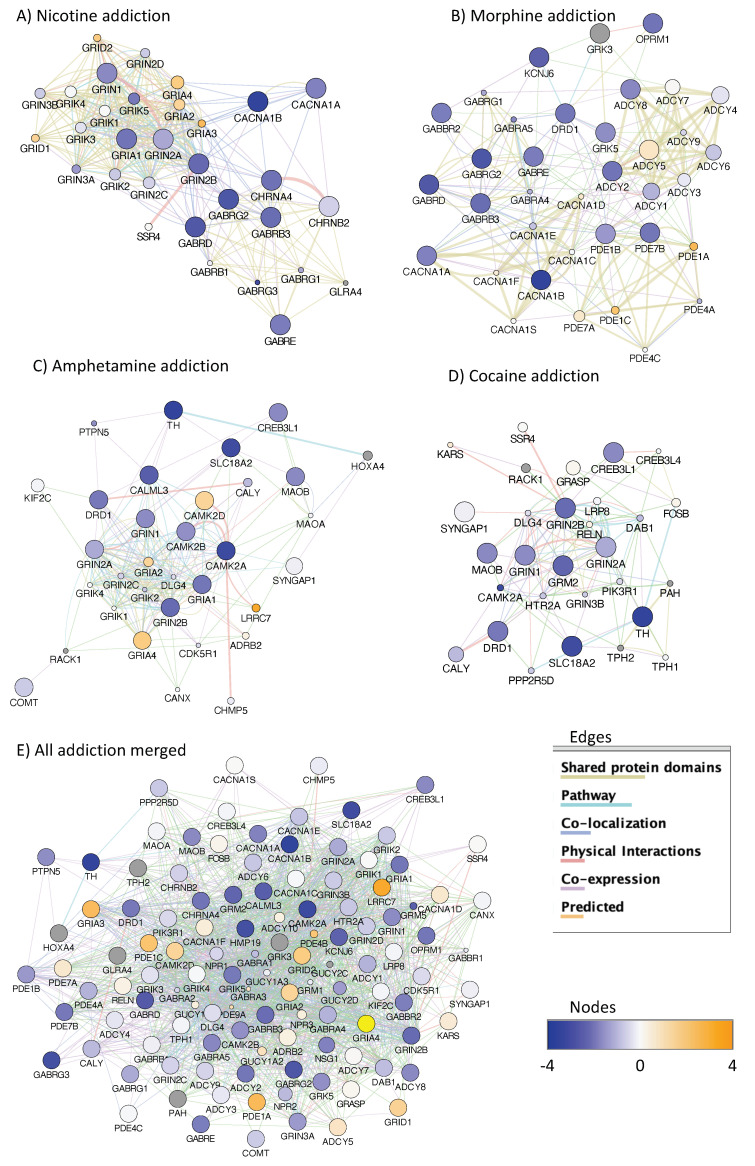
Gene clusters annotated to drug addiction pathways in HIV+ prefrontal cortex and the effect of Meth use. The analysis of genes with H3K4me3 active regions, including in-gene, downstream distal and proximal, as well as upstream has resulted in gene clusters annotated to addiction. Genes overrepresented in pathways associated to (**A**) Nicotine addiction (*p* = 2.2 × 10^−5^, Benjamini = 7.6 × 10^−4^), (**B**) Morphine addiction (*p* = 1.9 × 10^−5^, Benjamini = 1.3 × 10^−3^), (**C**) Amphetamine addiction (*p* = 1.4 × 10^−4^, Benjamini = 6.1 × 10^−3^) and (**D**) Cocaine addiction (*p* = 1.1 × 10^−3^, Benjamini = 3 × 10^−2^). (**E**) These clusters showed a 56% overlap and 100% connectivity in merged features. Node sizes represent connection scores. Blue colors in nodes indicate decrease and orange indicates increase in H3K4me3 peak signals in averaged HIV+Meth+ compared to HIV+Meth- prefrontal cortex specimens. Gray nodes indicate genes in the pathway for which we have not detected any H3K4me3 peak signal. Edge colors represent interaction criteria in connectors as defined in the legend.

**Figure 11 viruses-13-00544-f011:**
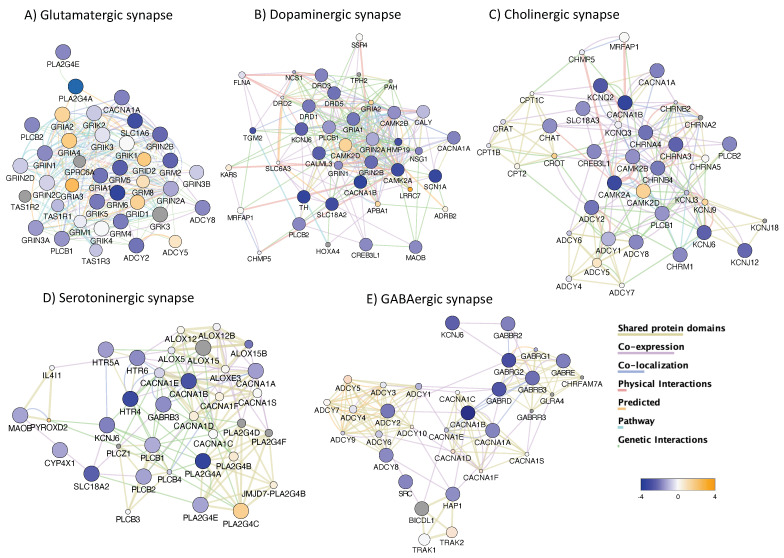
Gene clusters annotated to neurological synapses in HIV+ prefrontal cortex and the effect of Meth use. The analysis of genes with H3K4me3 active regions, including in-gene, downstream distal and proximal, as well as upstream, has resulted in gene clusters annotated to neurological synapses and function. Genes overrepresented in pathways associated to (**A**) Glutamatergic synapse (*p* = 5.6 × 10^−5^, Benjamini = 1.310^−3^), (**B**) Dopaminergic synapse (*p* = 2.2 × 10^−5^, Benjamini = 7.6 × 10^−4^), (**C**) Cholinergic synapse (*p* = 3.9 × 10^−5^), Benjamini = 1.2 × 10^−3^), (**D**) Serotoninergic synapse (*p* = 1.8 × 10^−3^, Benjamini = 1.6 × 10^−2^) and (**E**) GABAergic synapse (*p* = 3.6 × 10^−4^, Benjamini = 3.2 × 10^−2^).

**Figure 12 viruses-13-00544-f012:**
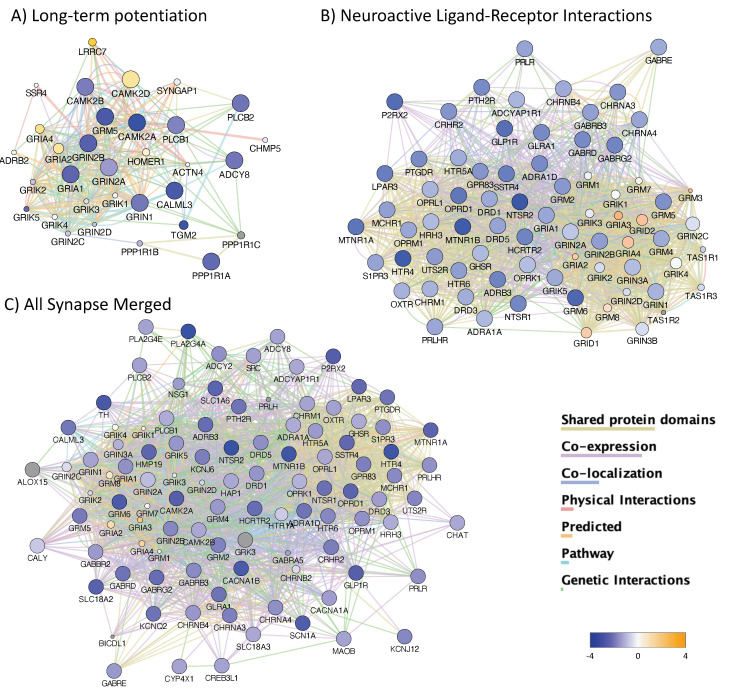
Gene clusters annotated to neurological functions. (**A**) Long-term potentiation (*p* = 1.4 × 10^−4^, Benjamini = 6.2 × 10^−3^). (**B**) Neuroactive ligand-receptor interactions (*p* = 6.6 × 10^−18^, Benjamini = 3.3 × 10^−16^). (**C**) Merged synaptic clusters indicating redundancy and common interactions. Clusters in Figure 11 and Figure 12 showed a 52.7% overlap and 100% connectivity in merged features. Node sizes represent connection scores. Blue colors indicate decrease and orange indicates increase in H3K4me3 peak signals in averaged HIV+Meth+ compared to HIV+Meth- prefrontal cortex specimens. Gray nodes indicate genes in the pathway for which we have not detected any H3K4me4 peak signal.

**Table 1 viruses-13-00544-t001:** Sequencing statistics by group.

	Average HIV+Meth- H3K4me3	Average HIV+Meth+ H3K4me3	Pooled Input hg38
Total number of reads	39,365,646	40,840,923	42,982,011
Total number of alignments (hg38)	35,790,215	38,438,156	39,813,903
Unique alignments (-q 25)	33,267,427	35,878,237	34,930,380
Unique alignments (without duplicates)	23,200,850	26,581,519	32,071,577

**Table 2 viruses-13-00544-t002:** Specimen subject groups, viral load at CD4 nadir, ART prescribed at any given time and RNA integrity number (RIN) following extraction.

Group	Case#	Plasma VL at CD4 nadir	ART History	Frontal Cortex RNA RIN
HIV+Meth-	4067	5.12	ZDV + 3TC, LPV + RTV, TFV	6.6
HIV+Meth-	4077	2.6	ATV, FTC, RTV, TFV, TFV + FTC	5.7
HIV+Meth-	4083	3.35	ZDV + 3TC, RFV, ATV, RTV, TFV/FTC	6.8
HIV+Meth+	1163	5.57	DRV, RTV, TFV + FTC	4.6
HIV+Meth+	2074	4.57	3TC, D4T, IDV, NVP, RTV, ABC, LPV + RTV, NFV, DDI, TFV	5.2
HIV+Meth+	4167	6.28	LPV + RTV, TFV + FTC	6.3

Legend: ART–Abbreviations: ZDV—zidovudine; 3TC—lamivudine; LPV—lopinavir; ATV—atazanavir; FTC—emtriditabine; RTV—ritonavir; TFV—tenofovir; DRV—darunavir; D4T—stavudine; IDV—indinavir; NVP—nevirapine; ABV—abacavir; NFV—nelfinavir; DDI—didanosine.

**Table 3 viruses-13-00544-t003:** List of 10 most frequent transcription factor binding motifs identified in +/−200 bp from and within H3K4me3 intervals in HIV+ Meth- and Meth+ specimens. The most frequent consensus sequences are presented, along with the average number of binding sites per interval sequence throughout the genome.

Matrix	Factor	Consensus Sequence	Classification	Average #Sites Interval Sequence
V$SP1_13	Sp1	gcggctctgcggGGCGGggcgggg	ZFC2H2	4.33
V$P53_03	P53	cgACATGGacacacatgggt	P53	3.33
V$ETS2_06	c-Ets-2	ggCCGGAgaggctgcccctt	ETS	2.85
V$JUNDFRA2_01	JUND:FRA2	aaTGACTcaa	BZIP	2.5
V$ASCL1_03	MASH-1	cgCAGCTgcc	BHLH	2.5
V$NR3C1_13	GR	aacacaataTGTACa	ZFC4-NR	2.33
V$CREBP1_01	ATF-2	tgaCGTCA	BZIP	2.29
V$AP2ALPHA_03	AP2aA	cgCGCCCccggctct	BHSH	2.27
V$MYCMAX_03	cMyc:Max	agttatgcACGTGtgtacca	BHLH	2.15
V$HOXA5_03	HOXA5	AATTAgtg	HOX	2

## Data Availability

All data relevant to this manuscript is available in Appendix A.

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
