# Peer review of "Detection of H3K4me3 Identifies NeuroHIV Signatures, Genomic Effects of Methamphetamine and Addiction Pathways in Postmortem HIV+ Brain Specimens that Are Not Amenable to Transcriptome Analysis"

_viruses, 2021, doi:10.3390/v13040544_

Round 1

Reviewer 1 Report

The manuscript is very thorough and very well-written. The experiments are well-panned and executed. The results are clear and presented in concise manner. The are presented very methodically, and appropriate methods were employed to answer a specific questions. The conclusions are also drawn appropriately based on the results obtained. Although there a couple of limitations, e.g. small sample size, the manuscript is thorough and informative enough to warrant publication.

This reviewer has three minor comments.

  1. A separate statistical analysis section should be included with the methods section.
  2. Figure 4A does not appear to indicate which bar is which. It appears to be three different time point. However, its not mentioned in the figure legend.
  3.  Since there is some correlation made based on the epigenetic changes with meth in HIV with other pathways that are related to other drugs of abuse e.g. nicotine, its important to include some discussion regarding this. 

Author Response

Responses to Reviewer 1:

  1. A separate statistical analysis section should be included with the methods section.

We have previously provided a statistical methods session, item 2.3 in Methods, which is applicable to ChiP-qPCR, and therefore follows the description of those methods. Other statistical methods that are part of the systems analysis are embedded and automatically applied to the data within the applications, and therefore we opted to described these methods as part of the analysis flow, rather than separating into a section, which would increase the complexity and to some extent introduce an element of redundancy.

  1. Figure 4A does not appear to indicate which bar is which. It appears to be three different time point. However, it’s not mentioned in the figure legend.

Thank you for pointing it out. It was in fact in the legend, but we agree that this was a confusing option. In the revised version, we have substituted the figure to incorporate a legend to the figure, which referred to the control non-transcribed gene island in the human chromosome 12, and the 2 favorite housekeeping genes, ACTB and GAPH. Please, notice that figure 4 has been replaced.

  1.  Since there is some correlation made based on the epigenetic changes with meth in HIV with other pathways that are related to other drugs of abuse e.g. nicotine, it’s important to include some discussion regarding this. 

This is a great suggestion. In the revised version, we expanded our comments in the discussion regarding the implications of findings related to a diversity of pathways involved in other drugs of abuse in addition to amphetamines, keeping in mind our limited knowledge about the subjects’ history. It is a fact that polysubstance use is prevalent and may be a factor in some of the specimens, but it is also important to acknowledge that molecular mechanisms involved in addiction have commonalities between different addictive drugs. This is evident in the overlap between all those pathways. In any case, it is worthy it a comment as suggested.

Reviewer 2 Report

This study from the Marcondes lab focuses on maximizing our capacity to analyze changes in gene expression in human tissue sections by examining the ability to use epigenetic markers to examine gene expression changes in poorly preserved human tissue sections. Specifically, they examined the use of two epigenetic marks, H3K4me3 and H3K27me3, that signal increased transcriptional activity at the genes they are associated with. This paper examines how detection of these marks is affected in tissues with lower quality RNA, and then used this optimized analysis to test whether the analysis of epigenetic data identifies viral, inflammatory, and neurological pathways in HIV+ prefrontal cortex tissue. The specifically examine tissue from HIV-infected Meth-users and non-Meth users, defining differences in addiction pathways. The data from this study is very useful, as they define the time course of RNA degradation in non-preserved tissue, using mouse brains to show that lack of preservation for 24hrs post-mortem leads to severe RNA degradation that would lead to inaccurate transcriptional data, but that epigenetic marks on DNA are still stable up to 96 hours. Using these timelines, the manuscript indicates that analysis of epigenetic marks could provide information about gene transcription patterns which would otherwise be lost due to poor RNA quality. As human postmortem tissue, particularly tissue from HIV+ infected individuals, is a scarce resource, these data provide a useful mechanism by which to maximize the utility of these tissues. This could expand the analysis of specimens in tissue repositories and enhance the investigation of global changes associated with HIV infection and neurological outcomes, as well as pathways involved in addiction, in a larger number of specimens.

            The research in this study was well done, and the results of the analysis of the potential of epigenetic marks to reveal information in low quality RNA was strong. Further, the use of this analysis to examine the changes in gene activity in Meth users and non-users infected with HIV is provides some useful information and adds to the existing literature from this and other labs in this area. Unfortunately, much of this information is difficult to access and understand due to the way the paper is written, principally the lack of explanation and discussion of the genomic analyses and the implications of those findings. The paper needs a substantial expansion of the description of the experiments, tables and readouts describing the data. For example, all of figure 6 is explained between lines 317 and 323, but this explanation is not sufficient to explain what the different regions (gene bodies vs. merged peak regions vs. promoters) are, how they are being compared, why the heatmaps have different sizes x-axes, what the distinct 5 clusters are in each sub panel and overall, what this figure is telling us. This problem is prevalent in many of the figures, such as Figure 7, in which the explanation in lines 342 – 344 and in the legend is inadequate and confusing. This is particularly true for individuals who are not well versed in genomic analyses or used to looking at these types of figures. In addition, the broader meaning of these data, particularly those evaluating the differences in gene expression   and of the meaning of these data, both in terms of their meaning within this study and in the context of the broader HIV / drug abuse literature. Figures 8 – 11 and the discussion spend very little time explaining why the gene clusters and the colors on the particular genes in these clusters are important to our understanding of substance abuse in HIV. While it seems that the major point of the paper is to show that epigenetic marks are a viable analysis technique, something the paper does well, the amount of time and real-estate devoted to the results of the transcriptional analyses necessitate a more substantive discussion of these results. The manuscript also needs syntactical and grammatical editing. Some specific comments giving examples of these issues are expanded on below.

  1. In lines 85-87, are there references that can be included for using ACTB and GAPDH in similar assays?

  1. The first two paragraphs of the Results section (lines 208 – 221) are quite redundant in relation to what is described in the introduction and discussion and could be omitted or shortened considerably.

  1. What are PCDH and A3C4 in Figure 1?

  1. In figure 1, the lack of explanation of the histograms and the meaning of the specific peaks undersells the effect of time on RIN in this data. For instance, the change in the scale on the Y axis represents a substantial shift in the amount of viable RNA, but that is very difficult to tell the way this is displayed. Also, please define the units FU for this scale in the legend.

  1. It is not clear what the Y axis is in Figure 2, is it transformed? And the binding events for Untr6 are significantly lower compared to ACTB and GAPDH, but is it really a negative control? Are there references to other studies that make comparisons in this way? The significance bars are confusing and could be represented in a cleaner way, maybe a different color or symbol for different comparisons? Also, to get a sense of the variability in the results obtained looking at different marks the individual data points from all of the tissues, i.e., what is the N for each column. These comments also apply to figure 4A.

  1. It would be nice to be able to compare the binding events per 1000 cells in Figures 2 and 4A, to get a sense of how the findings in terms of time before preservation apply to the human tissues.

  1. Was the RNA scope in Figure 3 quantified?

  1. In figure 4A, it seems like the meth+ tissues had higher binding events per 1000 cells than the meth- tissues. Was this simply due to the quality of the tissue, or was there a substantial difference here, potentially due to differences in the number of H3K4me3 marks with Meth- / + tissues?

  1. More explanation is needed for the results represented in Figure 4B? For example, explanation of different negative controls (Utr6 vs Utr12) between the mouse and human studies, even though the same positive control genes were used? Or explanation of what hg38 is and why there are no peaks on RNA from this sample?

  1. The Venn diagram in figure 5 misrepresents the amount of overlap vs. the number of separated genes (the overlap section should be much larger), and the pie graphs in this figure are very hard to compare between conditions, making it difficult to determine whether there are any changes in the HIV+Meth- and HIV+ Meth+ conditions. Also, the text on the hg38 pie chart needs to be fixed and it is necessary to define what hg38 is.

  1. Figures 8 – 11 are very difficult to access – particularly Figure 8 - as the text is compressed, explanations are inadequate, and the picture quality is not consistent across the figures. For Figure 10 and 11, the merged pathway images in particular are probably unnecessary and would be more beneficial if just described in the text. This information could be better visualized as a table, or a heat map, or perhaps a more stringent cutoff to look at the gene clusters as images.

  1. In Figure 10, it is not clear what the connection scores represented by the node sizes are, nor it is clear what is meant by edges – if that means different colored edges, they should be thickened because they are not identifiable. Also, the H3K4me4 should be corrected in the legend.

Author Response

Responses to Reviewer 2:

  1. Unfortunately, much of this information is difficult to access and understand due to the way the paper is written, principally the lack of explanation and discussion of the genomic analyses and the implications of those findings. The paper needs a substantial expansion of the description of the experiments, tables and readouts describing the data.

Thank you for this comment, we have extensively revised the text following your detailed suggestions. Sometimes we fail to see our lack of clarity due to our own familiarity with the subject. We appreciate the opportunity to amend this problem and improve our manuscript.

  1. For example, all of figure 6 is explained between lines 317 and 323, but this explanation is not sufficient to explain what the different regions (gene bodies vs. merged peak regions vs. promoters) are, how they are being compared, why the heatmaps have different sizes x-axes, what the distinct 5 clusters are in each sub panel and overall, what this figure is telling us.

We have now expanded the explanation of the figure.

  1. This problem is prevalent in many of the figures, such as Figure 7, in which the explanation in lines 342 – 344 and in the legend is inadequate and confusing. This is particularly true for individuals who are not well versed in genomic analyses or used to looking at these types of figures. In addition, the broader meaning of these data, particularly those evaluating the differences in gene expression and of the meaning of these data, both in terms of their meaning within this study and in the context of the broader HIV / drug abuse literature.

We have expanded the description of Figure 7, not only in results, but also in discussion, as a reference to apoptosis as an important biological process in Meth users.

  1. Figures 8 – 11 and the discussion spend very little time explaining why the gene clusters and the colors on the particular genes in these clusters are important to our understanding of substance abuse in HIV. While it seems that the major point of the paper is to show that epigenetic marks are a viable analysis technique, something the paper does well, the amount of time and real-estate devoted to the results of the transcriptional analyses necessitate a more substantive discussion of these results.

The reviewer is completely right and the point was well taken. In this revised version, we provided a cautious interpretation of the data in perspective with drug use in the context of HIV, including an extended description of gene relationships indicating that processes occur in concert. Still, given that the specimens do not provide the opportunity to validate via a parallel transcriptome analysis, we acknowledge the limitations that impede a rigorous interpretation.

  1. The manuscript also needs syntactical and grammatical editing. Some specific comments giving examples of these issues are expanded on below.

We really appreciate. I would like to highlight that the manuscript has been subjected to editor services that are internally offered by our institute.

  1. In lines 85-87, are there references that can be included for using ACTB and GAPDH in similar assays?

Thank you for this comment. These genes are standards, housekeeping genes that are always constitutively transcribed, and thus perfect positive controls for ChIP quality. We have now added a rationale sentence, as well as a reference.

  1. The first two paragraphs of the Results section (lines 208 – 221) are quite redundant in relation to what is described in the introduction and discussion and could be omitted or shortened considerably.

Thank you for pointing out. We have revised and merged these paragraphs, as a brief overview of the strategy.

  1. What are PCDH and A3C4 in Figure 1?

Thank you for asking. These are RNA quality references that are used in the Scripps Research Core Facility. We have included in the legend.

  1. In figure 1, the lack of explanation of the histograms and the meaning of the specific peaks undersells the effect of time on RIN in this data. For instance, the change in the scale on the Y axis represents a substantial shift in the amount of viable RNA, but that is very difficult to tell the way this is displayed. Also, please define the units FU for this scale in the legend.

This is an excellent suggestion, and very fine observation. We have improved our explanation, both in the Results text and in the legend.

  1. It is not clear what the Y axis is in Figure 2, is it transformed? And the binding events for Untr6 are significantly lower compared to ACTB and GAPDH, but is it really a negative control? Are there references to other studies that make comparisons in this way? The significance bars are confusing and could be represented in a cleaner way, maybe a different color or symbol for different comparisons? Also, to get a sense of the variability in the results obtained looking at different marks the individual data points from all of the tissues, i.e., what is the N for each column. These comments also apply to figure 4A.

Thank you for the comment. As mentioned before, we have added references on the method, which is standard. Untr6 for mouse and 12 for human are true negative controls, because these are what they call “untranslated genomic islands”. These regions are not annotated to any gene or regulatory sequence, and no non-coding DNA sequence functions have been so far attributed to them. Moreover, these are regions that do not map to any epigenetic or genomic activity. Why this is like this, I don’t know. But ACTB, GAPDH and the Untr “genes” are purchased as a set of control primers, specific for mouse or human (same genes but different primers), specifically to test ChIP assays. The reason why the scale is in log, is due to the huge difference in RNAPol frequency compared to enhancing events H3K4me3 and H3K27Ac. It is perhaps important to point that the frequency and fold enrichments were within the expected range for specimens at 0 and 6 hrs.

  1. It would be nice to be able to compare the binding events per 1000 cells in Figures 2 and 4A, to get a sense of how the findings in terms of time before preservation apply to the human tissues.

We agree that it could be a nice way to estimate postmortem interval. However, given that with the current human specimens we cannot systematically make a comparison, we will keep it as a suggestion for our next level.

  1. Was the RNA scope in Figure 3 quantified?

This is a fantastic question. We did not quantify for this study, for 2 reasons. 1) Because tissue quality is an issue. For instance, we see that virus is not detected in specimens that had high viral load, which again speaks to the problem of quality. 2) Because NNTC provided sections of one fragment. In order to safely quantify, we may need to sample deeper or other prefrontal fragments, and to draw a conclusion, in a larger number of specimen subjects, with preference to the ones that have good RNA quality in parallel frozen specimens. So, for now, within the scope of this study, we will use the RNA detection + or – as seen in the picture, as a variable that is potentially affected by RNA quality.  

  1. In figure 4A, it seems like the meth+ tissues had higher binding events per 1000 cells than the meth- tissues. Was this simply due to the quality of the tissue, or was there a substantial difference here, potentially due to differences in the number of H3K4me3 marks with Meth- / + tissues?

That is a very good question. We have extensively tested the quality of the chromatin preparations and documented that the distribution of tags was homogeneous in both groups making us comfortable about comparisons. However, differences may still occur, and they could be due to either one of the suggested reasons or both.

  1. More explanation is needed for the results represented in Figure 4B? For example, explanation of different negative controls (Utr6 vs Utr12) between the mouse and human studies, even though the same positive control genes were used? Or explanation of what hg38 is and why there are no peaks on RNA from this sample?

Thank you for pointing to this problem. We have extended our explanations as mentioned above.

  1. The Venn diagram in figure 5 misrepresents the amount of overlap vs. the number of separated genes (the overlap section should be much larger), and the pie graphs in this figure are very hard to compare between conditions, making it difficult to determine whether there are any changes in the HIV+Meth- and HIV+ Meth+ conditions. Also, the text on the hg38 pie chart needs to be fixed and it is necessary to define what hg38 is.

We have fixed the text according to these suggestions.

  1. Figures 8 – 11 are very difficult to access – particularly Figure 8 - as the text is compressed, explanations are inadequate, and the picture quality is not consistent across the figures. For Figure 10 and 11, the merged pathway images in particular are probably unnecessary and would be more beneficial if just described in the text. This information could be better visualized as a table, or a heat map, or perhaps a more stringent cutoff to look at the gene clusters as images.

Thank you for these suggestions. We are providing additional supplementary data with these merged genes in excel format. In addition, we believe that part of the problem was that Figure 11 was busy, and clusters were small as a result. We find the merging important, because the redundancy of pathways may be an explanation for identifying gene clusters annotated for instance to other drugs of abuse. Thus, we opted to split Figure 11 into 2 Figures. This way, the clusters are larger and easier to see, including the lines and gene numbers.

  1. In Figure 10, it is not clear what the connection scores represented by the node sizes are, nor it is clear what is meant by edges – if that means different colored edges, they should be thickened because they are not identifiable. Also, the H3K4me4 should be corrected in the legend.

We have clarified the meaning of scores in the text and revised edge thickness. Legend for Figure 10 was fixed.

Again, we would like to thank the reviewers for the detailed comments. This was incredibly helpful, and we believe the manuscript has improved a lot in clarity to readers that are not familiar with systems biology. This will help us reach a broader number of people to communicate this relevant problem, and a potential solution.

Please, don’t hesitate to contact us again if needed.
